# Contactless Measurement of Vital Signs Using Thermal and RGB Cameras: A Study of COVID 19-Related Health Monitoring

**DOI:** 10.3390/s22020627

**Published:** 2022-01-14

**Authors:** Fan Yang, Shan He, Siddharth Sadanand, Aroon Yusuf, Miodrag Bolic

**Affiliations:** 1Health Devices Research Group (HDRG), School of Electrical Engineering and Computer Science, University of Ottawa, Ottawa, ON K1N 6N5, Canada; she014@uottawa.ca (S.H.); miodrag.bolic@uottawa.ca (M.B.); 2Institute for Biomedical Engineering, Science and Technology (iBEST), St. Michael’s Hospital, Ryerson University, Toronto, ON M5B 1T8, Canada; siddharth.n.v.sadanand@gmail.com; 3WelChek Inc., Mississauga, ON L4W 4Y8, Canada; dra.yusuf@gmail.com

**Keywords:** vital signs, heart rate (HR), respiration rate (RR), body temperature (BT), face detection, image processing, signal processing

## Abstract

In this study, a contactless vital signs monitoring system was proposed, which can measure body temperature (BT), heart rate (HR) and respiration rate (RR) for people with and without face masks using a thermal and an RGB camera. The convolution neural network (CNN) based face detector was applied and three regions of interest (ROIs) were located based on facial landmarks for vital sign estimation. Ten healthy subjects from a variety of ethnic backgrounds with skin colors from pale white to darker brown participated in several different experiments. The absolute error (AE) between the estimated HR using the proposed method and the reference HR from all experiments is 2.70±2.28 beats/min (mean ± std), and the AE between the estimated RR and the reference RR from all experiments is 1.47±1.33 breaths/min (mean ± std) at a distance of 0.6–1.2 m.

## 1. Introduction

With the outbreak of the COVID-19 pandemic, the worldwide healthcare systems are facing challenges that have never been met before. Most common symptoms of COVID-19 include fever, dry cough, and shortness of breath or difficulty breathing. Some researchers have found that COVID-19 can cause unexplained tachycardia (rapid heart rate) [1]. It is important and useful to monitor the vital signs of people before they get a diagnosis from the hospital as well as to track the vital signs during their recovery. The vital signs, such as body temperature (BT), respiratory rate (RR) and heart rate (HR), could be used as evidence for prescreening suspected COVID-19 cases. In order to assist in detecting COVID-19 symptoms, we developed a contactless vital sign monitoring system, which can detect the BT, HR and RR at a safe social distance (60–120 cm). The proposed system and algorithms can be potentially applied for quick vital signs measurement at the entry of buildings, such as shopping malls, schools and hospitals.

Compared to previous work [2,3,4], our method improves the contactless vital signs measurement range up to 2 m for RR estimation and 1.2 m for HR estimation while preserving the high estimation accuracy. Besides, our proposed approach addresses the challenge when the subject is wearing the mask and the face is partly occluded using the CNN based face detector. In addition, the face detector in our method enables us to measure multiple subjects’ vital signs such as RR and HR at the same time, which, to the best of our knowledge, has never been studied before. To summarize, our method makes the following contributions:We present a system that estimates multiple subjects’ vital signs including HR and RR using thermal and RGB cameras. To the best of our knowledge, it is the first study that includes different face masks in contactless RR estimation and our results indicate that the proposed system is feasible for COVID 19-related applications;We propose signal processing algorithms that estimate the HR and RR of multiple subjects under different conditions. Examples of novel approaches include increasing the contrast of the thermal images to improve the SNR of the extracted signal for RR estimation as well as a sequence of steps including independent component analysis (ICA) and empirical mode decomposition (EMD) to enhance heart rate estimation accuracy from RGB frames. Robustness is improved by performing a signal quality assessment of the physiological signals and detecting the deviation in the orientation of the head from the direction towards the camera. By applying the proposed approaches, the system can provide accurate HR and RR estimations with normal indoor illuminations and for subjects with different skin tones;Our work addresses some of the issues reported in other works such as the small distance required between the cameras and the subjects and the need to have a large portion of the face exposed to the camera. Therefore, our system is robust at larger distances, and can simultaneously estimate the vital signs of two people whose faces might be partially covered with face masks or not pointed directly towards the cameras.

## 2. Related Works

Conventionally, contact based devices are widely used in hospitals and nursing homes for the measurement of vital signs. For example, electrocardiography (ECG) and photoplethysmography (PPG) sensors are usually used in HR monitoring [5] and the breathing belt is used to measure the RR of the subject. However, these devices need to be attached to the subjects’ body thereby constraining or causing them discomfort and thus potentially affecting the measurement. It is also hard to make many people wear this kind of sensor and collect their data when they are in public places.

To resolve these problems, contactless monitoring systems have been designed and applied in the aforementioned scenario. Contactless monitoring systems rely on a number of sensors such as radars, sonars, different kind of camera and so on. In this paper, we will limit our analysis to camera-based systems. Xiaobai Li and Jie Chen et al. proposed a method that can measure HR remotely from face videos [2]. After that, many researchers used the green channel of face images collected from a webcam or smartphone to measure the heart rate [6,7]. On the other side, with the decline of the cost and size of optical imaging devices such as infrared cameras and thermal cameras, the respiratory signal can also be measured by remote monitoring [8,9,10]. Youngjun Cho et al. proposed a respiratory rate tracking method based on mobile thermal imaging [3]. One low-cost and small-sized thermal camera connected to a smartphone was used as a thermographic system to detect the temperature changes of the nostril area caused by the inhaling and exhaling. There have been some other works [4,11,12] related to monitoring the respiratory signal using the thermal camera. However, the detection distance between the human face and the thermal camera is limited to less than 50 cm in Cho’s work [3] and the estimation error increases with the distance between the subject and the camera. In addition, some face detection algorithms [3,4,11] perform poorly in detecting occluded faces if the subjects are wearing face masks and make it improper for COVID symptoms’ detection as a face mask is always mandatory in public places.

In our task of monitoring a subject’s HR and RR using cameras, the subject’s face should be detected from the video frames where the face detection and tracking algorithms need to be developed and applied. The Kanade–Lucas–Tomasi (KLT) feature tracker originally proposed in 1991 is generally used to track the faces and the region of interest (ROI) in videos [5]. In addition, there are many face detectors in OpenCV and Matlab toolboxes such as the Viola–Jones face detector [13] and the Discriminative Response Map Fitting (DRMF) detector [14]. However, these detectors have a problem handling the occluded faces so they cannot be used in our scenario. Face detection also receives attention in the deep learning community where many convolutional neural network frameworks have been developed in recent years to address the challenges. Multi-task Cascaded Convolutional Networks (MTCNN), proposed by Kaipeng Zhang et al. in 2016, which adopt a cascaded structure of deep convolutional networks to detect the face and landmarks [15], were widely used in the task of detecting faces. However, MTCNN cannot resolve the problem when the face is covered with a mask. The Single Shot Scale-invariant Face Detector (S3FD) [16] proposed in 2017 also does not perform well in occluded face detection. PyramidBox, proposed in 2018 [17], is capable of handling the hard face detection problems such as blurred and partially occluded faces in an uncontrolled environment. Though PyramidBox has already been applied to detecting faces occluded by the mask in the application of respiratory infections detection using RGB-Infrared sensors [18], it cannot be applied to scenarios where the facial landmarks or ROI detection are required. Besides the face detection, fitting methods for face alignment [19] are also significant in our task, because we need to detect the interested region based on facial landmarks. However, this landmarks detection relies on complete faces without occlusion.

## 3. Materials and Protocols

### 3.1. Data Acquisition System

In this study, a thermal camera (Rakinda Technology, FT20) and an RGB webcam (Logitech©, C270) were employed to collect thermal and RGB videos from human subjects. The thermal camera operates at 25 fps and its resolution is 256 × 192, the RGB camera operates at 30 fps and its resolution is 640 × 480. These cameras are placed side by side horizontally on a pole and the height is 1.6 m. An LED camera light (Neewer, model 176 LED) was placed close to the cameras to enhance illuminations on subjects’ faces and to avoid shadows (normal indoor illumination level). The setup of the devices is shown in Figure 1. To evaluate the performance of the proposed system in vital sign estimation, an optical PPG sensor (Sparkfun, heart rate sensor), a force-based chest respiration belt (Vernier, Go Direct) and a handheld infrared forehead thermometer were used as reference measurements of heart rate, respiration rate and body temperature, respectively.

A desktop (Intel Core i9, Nvidia GeForce RTX 2080 Ti, 64 GB RAM, Ubuntu 20 OS) was used for video acquisitions from the thermal and RGB cameras, reference signal acquisition, visualization, data storage and analysis. Face detection and tracking, signal processing and the proposed algorithms were developed using Python 3.7.

### 3.2. Experimental Protocols

In this study, ten healthy subjects (seven males, three females, 24–35 years old) participated in several different experiments. The subjects came from a variety of ethnic backgrounds with skin colors from pale white to darker brown. The protocol was approved by the University of Ottawa Research Ethics Board.

One subject (male, 27 years) participated in RR estimation with different face masks at different detection ranges. At each detection range (75 cm, 120 cm and 200 cm), the subject was asked to stand still in front of the thermal and RGB cameras without face mask and with three different face masks (medical/cloth, N90, N95) for two minutes respectively. The subject performed normal breaths during the test. These experiments aim to test the performance of RR monitoring using a thermal camera when a subject has a face cover.

Ten healthy subjects (seven males, three females, 24–35 years old) participated in one-subject vital sign estimation experiments. Each subject was asked to keep stationary and breathe normally in front of the cameras at four different detection ranges (60 cm, 80 cm, 100 cm and 120 cm) separately. At each distance, simultaneous thermal and RGB videos were collected from the subject with and without a face mask (medical/cloth mask) for two minutes respectively. The simultaneous reference respiration waveform was collected from the subject’s chest and the reference PPG signal was collected from the subject’s fingertip. The reference body temperature was measured by the infrared thermometer from the subject’s forehead every 30 s.

Six subjects (four males, two females, 24–28 years old) participated in two-subjects vital sign estimation experiments. In each experiment, two subjects were asked to keep stationary and breathe normally in front of the cameras at different distances (70 cm, 100 cm) for two minutes respectively. Subjects were not wearing face masks in Experiments 1–3 and they were wearing face masks in Experiments 4–5 as demonstrated in Section 5.2.3. The reference respiration waveforms were collected from subjects’ chests and the reference PPG signals were collected from the subject’s fingertips simultaneously.

## 4. Methods

The steps of remote estimation of heart rate, respiration rate and body temperature are shown in Figure 2. In our system, RGB and thermal videos are collected by RGB and thermal cameras simultaneously, and the sampling rate of these two cameras is 25 frames per second (fps). The RGB camera is used to detect and track human subjects’ faces, and thus locate three different regions of interest (ROIs) for vital signs estimation. The forehead area from the RGB video sequence is extracted to estimate HR. In addition, with the help of image alignment process, a point from the forehead area and the nostril area can be located on the simultaneous thermal video frames for BT and RR estimation.

### 4.1. Face Detection

We applied the RetinaFace [20] framework to detect faces from the RGB video frames. RetinaFace was proposed by Deng et al. and ranks second in the competition of WIDER FACE (Hard). The WIDER FACE dataset [21] is a face detection benchmark dataset with 32,203 images and 393,703 labeled faces which has a high degree of variability in scale, pose and occlusion. RetinaFace is a practical single-stage deep learning based face detector working on each frame of the video. Consequently, we chose it because only the RetinaFace detector can detect both face and facial landmarks with the confidence score over 0.9 comparing to some widely used face detectors including MTCNN, S3FD and PyramidBox. An example of face and facial landmarks detection using these face detectors when a subject was wearing a face mask is illustrated in Figure 3.

Additionally, RetinaFace can detect multiple faces from one image as illustrated in Figure 4, which makes it possible for our system to detect multiple subjects’ vital signs even when their faces are partially occluded by face masks at the same time.

### 4.2. Regions of Interest (ROIs)

In this study, the proposed system aims to monitor three vital signs (BT, HR and RR) remotely. Therefore after detecting subject’s face, three different regions of interest (ROIs) needs to be located from RGB or thermal video frames. The locations of the ROIs are illustrated in Figure 5.

#### 4.2.1. ROI for BT Estimation

Like most of the non-contact thermometers measurements, a single point ($ROI_{BT}$) on the forehead is identified as the interested point $\left(T_{x},T_{y}\right)$ for body temperature estimation in our system. After detecting the bounding box of the face and locating the positions of the eyes, $T_{y}$ is defined as $H_{1}/3$ above the middle point of the left and right eyes, where $H_{1}$ is the difference between the upper edge of the bounding box and the middle point of eyes on y-axis. $T_{x}$ equals the mean value of *x* coordinates of the left and right eyes. A demonstration of locating $ROI_{BT}$ using the bounding box and facial landmarks is shown in Figure 6a.

#### 4.2.2. ROI for RR Estimation

In this study, a nostril is determined as the $ROI_{RR}$ for RR estimation as shown in Figure 5b. Warm air from inside the lungs is released through the respiratory system and it increases the temperature in the nasal region during exhalation, whereas cool air from the external environment is breathed in and it lowers the temperature in the nasal region during inhalation. Therefore, the respiration waveform can be obtained by using an infrared thermal camera to measure such nasal-region temperature changes associated with respiration. Consequently, the nostril areas are chosen as an ROI for respiratory signal detection. The nose tip landmark is chosen as the center-point of the nostril area. The width of $ROI_{RR}$ is defined as 90 percent of the mouth width $W_{2}$, and the height of $ROI_{RR}$ is 60 percent of the vertical difference ($H_{2}$) between the mouth middle point and the nose tip. An illustration of $ROI_{RR}$ can be found in Figure 6a.

#### 4.2.3. ROI for HR Estimation

Phenomena exploited in rBVP (remote blood volume pulse) for HR estimation are closely related to the cardiac cycle. During each cycle, blood is moved from the heart to the head through the carotid arteries. This periodic blood inflow affects both optical properties of facial skin and the mechanical movement of the head, enabling researchers to measure HR remotely [22].

Several different ROIs used for remote HR estimation are described in existing research. Some researchers used the bounding box of the face given by the face detection algorithm as the ROI [6,23,24,25,26]. Other researchers used the forehead area of the detected face as the ROI for HR estimation [6,23,27]. Through our experiments, the forehead area always performed better than the face area in HR estimation and the forehead will not be occluded by a face mask, therefore the forehead was determined as $ROI_{HR}$. The center point of the $ROI_{HR}$ is $ROI_{BT}$$\left(T_{x},T_{y}\right)$. The height of the $ROI_{HR}$ is 0.4 of the height of the bounding box and the width of the $ROI_{HR}$ is 0.5 of the width of the bounding box detected by the RetinaFace detector. A demonstration of $ROI_{HR}$ location can be found in Figure 6b.

### 4.3. Head Movement Detection

The facial landmarks are important for the division of the face and detection of our interested regions as illustrated in Figure 7a. To achieve accurate estimations of the vital signs, the subjects are supposed to keep stationary and look at the cameras during the data collection. However, there may be some scenarios where the subjects move their heads to other directions as shown in Figure 7b. If the movement is too large, we assume that it is not possible to reliable extract the vital signs and therefore, there is no longer need to extract the data from the ROIs. We calculate the ratio of the distance between the midpoint of the two detected eyes and the midpoint of the face along the *x* axis to the width of the face box:(1)$$Face_{deflection}=\frac{|T_{x}-Face_{x}|}{Face_{width}},$$
where $T_{x}$ represents the *x* coordinate of the midpoint of the left eye and the right eye, $Face_{x}$ represents the *x* coordinate of the midpoint of the face box. $Face_{width}$ is the width of the face box, and $Face_{deflection}$ denotes the ratio of the distance between the eyes’ midpoint *x* coordinate and the face midpoint *x* coordinate to the face width. The ratio $Face_{deflection}$ increases with the face moving away from the camera and the variation of the value describes the status of the subject’s face whether still or moving. We set the ratio $Face_{deflection}$ threshold to 0.17 when the face is not towards the camera as illustrated in Figure 7b and we do not need to process the data to estimate vital signs under this condition anymore. It is noted that we only consider horizontal face inclines rather than the vertical inclines because our device is mounted at a certain height and the subject is required to face the camera during the experiments.

### 4.4. Frame Registration

After locating the ROIs from the RGB frame, the $ROI_{RR}$ and $ROI_{BT}$ also needs to be located on the thermal image as the RR and BT estimations are based on temperature readings. The resolution of the thermal camera is also lower than the RGB camera, hence an image alignment method needs to be applied. In our system, two cameras are placed side by side in the same plane. Consequently, the alignment of frames from two cameras is simplified to an affine transformation problem. Affine transformation is a linear mapping method that preserves points, straight lines, and planes. Sets of parallel lines remain parallel after an affine transformation. We only consider translation and scaling in our two-camera system.

Transformation matrix *T* is defined as:(2)$$T=\left[\begin{array}{ccc} s_{x} & 0 & 0\\ 0 & s_{y} & 0\\ t_{x} & t_{y} & 1 \end{array}\right],$$
where *t${}_{x}$* specifies the displacement along the *x* axis, *t${}_{y}$* specifies the displacement along the *y* axis, *s${}_{x}$* specifies the scale factor along the *x* axis, and *s${}_{y}$* specifies the scale factor along the *y* axis.

Translation transformation is caused by the difference between the position of the two parallel cameras. Since the imaging planes of the RGB camera and the thermal camera are within the same plane, the displacements are along the *x* axis and the *y* axis. Scale transformation is caused by the focal length. The lens of the RGB camera has a shorter focal length compared to the lens of the thermal camera, and so the RGB camera captures a much wider field of view while producing a smaller picture. Figure 8a shows the original frame with 640×480 captured by the RGB camera, and Figure 8b,c show the synchronous and registered thermal frame and RGB frame with 256×192 using the transformation matrix *T*.

### 4.5. Vital Signs Estimation

#### 4.5.1. Body Temperature Measurement

Thermal cameras can measure a subject’s surface skin temperature without being physically close to the person being evaluated. The FT20 thermal imaging module provides us with 256×192 temperature values in each frame. Among them, the device outputs the data including the temperature of the center, the highest temperature and location, the lowest temperature and location, and the overall temperature in each pixel location within the frame. Consequently, we estimate the forehead skin temperature by directly extracting from the location of the forehead center point $\left(T_{x},T_{y}\right)$ in the thermal frame.

All of the 256×192 temperature values in one thermal frame are raw data generated by the thermal camera following the sequence of a pixel array. However, the raw temperature data are processed and transformed using the palette to be visible as pixels in the output thermal frame. Thus, the coordinate of the forehead center point $\left(T_{x},T_{y}\right)$ functions as a key of the dictionary to extract forehead temperature from the table of raw data.

The calibration of the thermal camera is done by using a black body device and has been tested by the camera manufacturer before the delivery. However, the thermal system measures surface skin temperature, which is usually lower than a temperature measured orally, and the experiments also show that the detection range and environmental temperature could have an impact on the measurement. So there are some difference between the measured value and the real body temperature. We also compare our temperature measurements with the handheld thermometer measurements, which only shows a subtle difference of less than 0.5 Celsius degrees when measuring the forehead skin temperature.

#### 4.5.2. Heart Rate Estimation

The steps of HR estimation proposed in this study are shown in Figure 9. After locating the $ROI_{HR}$ from an RGB video frame by frame, the raw color channel signals can be extracted by calculating the mean values of red, green and blue channels within $ROI_{HR}$. A 3rd order Butterworth bandpass filter with the passband of 0.8–2 Hz (corresponding to 48–120 beats/min) is applied on raw color signals for noise reduction.

Several different approaches have been considered and tested. The independent component analysis (ICA) method and the empirical mode decomposition (EMD) based signal selection method which perform well in HR estimation for subjects with different skin tones and in different illuminations are proposed for rBVP waveform recovery. The ICA, which was originally employed in [7], is widely used in vision-based HR estimation. It is a blind source separation technique used to separate multivariate signals from underlying sources. ICA can decompose the RGB signals into three independent source signals. Any of them can carry rBVP information. For $ROI_{HR}$ extracted at time point *t*, the denoised signals from red, green and blue channels are denoted as $x_{1}\left(t\right)$, $x_{2}\left(t\right)$ and $x_{3}\left(t\right)$ respectively. Then, these RGB traces are normalized as follows:(3)$$x_{i}^{\prime }\left(t\right)=\frac{x_{i}\left(t\right)-\mu_{i}}{\sigma_{i}},$$
where $\mu_{i}$ and $\sigma_{i}$ for i=1,2,3 are the mean and standard deviation of $x_{i}\left(t\right)$, respectively. The normalization transforms $x_{i}\left(t\right)$ to $x_{i}^{\prime }\left(t\right)$, which is zero-mean and has unit variance.

The normalized traces are then decomposed into three independent source signals using ICA. The joint approximate diagonalization of the eigenmatrices (JADE) algorithm developed by Cardoso [28] is applied, and the second component, which always contains the strongest plethysmographic signal, is selected as the desired source signal. Then the EMD, which can decompose the signal into a series of IMFs (Intrinsic Mode Functions), is applied to the extracted signal and the power of each IMF is calculated. The IMF with the maximum power is selected as the recovered rBVP signal for HR estimation.

The systolic peaks of the extracted rBVP signal were detected using a peak detection algorithm. Then the mean time interval $TI_{HR}$ between two consecutive systolic peaks was calculated. Therefore, the HR estimated using an RGB camera can be obtained as $HR_{RGB}=60/TI_{HR}$ (beats/min). An example of the extracted rBVP waveform using the proposed approaches and the simultaneous reference PPG waveform is shown in Figure 10b.

#### 4.5.3. Respiration Rate Estimation

After locating the nasal region from the thermal video, the respiratory signal is extracted by averaging the values of all pixels within $ROI_{nostril}$ frame by frame. However, the respiratory signal extracted directly from the original thermal frames has a small magnitude when the respiration-induced thermal variance is weak, for example, during shallow breathing or at a long detection distance. Consequently, we applied a histogram equalization to solve the weak signal problem. Here we use the Contrast Limited Adaptive Histogram Equalization (CLAHE) algorithm [29] to increase the contrast by spreading out the most frequent intensity values. As a result, we enhance the variation of ROI within thermal frames as illustrated in Figure 11 before we attempt to extract the respiratory signal.

Then, a 3rd order Butterworth bandpass filter with a lower cutoff frequency of 0.15 Hz and a higher cutoff frequency of 0.5 Hz (corresponding to 9–30 breaths/min) was applied to the extracted respiratory signal for noise removal.

The peaks of the extracted respiration signal were detected using a peak detection algorithm. Then the mean time interval $TI_{RR}$ between two consecutive peaks was calculated. Therefore, the RR estimated using a thermal camera can be obtained as $RR_{thermal}=60/TI_{RR}$ (breaths/min). An example of the extracted respiratory waveform using the proposed approaches and the simultaneous reference respiration waveform from a breath belt is shown in Figure 10a. It is worth mentioning that the reference breath belt is a force sensor based device and the force value increases during inhalation and decreases during exhalation; however, the temperature at the nostril area decreases during inhalation and increases during exhalation.

### 4.6. Signal Quality Evaluation

An ensemble averaging based signal quality assessment method was implemented to evaluate the stationarity of the extracted respiratory waveform and rBVP waveform. First, the peaks of the target signal are detected. The template window length is determined as the median value of the durations between two successive peaks. Individual pulses are then aligned with their peaks centered in the middle of the template window and these pulses are averaged to form the signal template. The Pearson correlation coefficients between each individual pulse and the signal template are calculated and their mean value is extracted as the signal quality index (SQI) of this signal segment. Finally, the signal segment will be used for RR and HR estimation if $SQI_{RR}>0.7$ and $SQI_{HR}>0.8$ respectively. An example of the proposed signal quality assessment method applied on a 15 s extracted rBVP signal is shown in Figure 12.

## 5. Results

### 5.1. Facial ROIs Detection

Our ROI detection method based on the RetinaFace detector, is deployed both on a desktop computer and an embedded device Nvidia Jetson Board for testing. The classification scores showing the probability of the detection of a human face that are tested on all of the four subjects at different positions with or without mask reach 99.95% or even higher using Retinaface detector. Besides, there are no difference between the subject wearing the mask and the one without mask when we evaluate the face classification probabilities. In addition, to evaluate the performance of facial landmarks detection which is the base of our facial ROI detection, we use the normalised mean error (NME) metric:(4)$$NME=\frac{1}{N}\sum_{i=1}^{N}\frac{\sqrt{\Delta x_{i}^{2}+\Delta y_{i}^{2}}}{d},$$
where *N* denotes the number of facial landmarks, which is five in our situation and *d* denotes the normalized distance. $\Delta x_{i}^{2}$ and $\Delta y_{i}^{2}$ are deviations between the *i*th predicted landmark and ground truth in *x* axis and *y* axis. We employ the face box size ($\sqrt{W\times H}$) as the *d* in the evaluation. We found that the average of NME is 1.5% when the subject is not wearing the mask, while the NME reaches 2.4% when the subject is wearing the mask.

In terms of execution time, we applied the whole RetinaFace detector on the embedded device NVIDIA Jetson TX2 board. Jetson TX2 is built around an NVIDIA Pascal™-family GPU and loaded with 8 GB of memory and 59.7 GB/s of memory bandwidth. The RetinaFace detector takes average 5 s to load the whole network model. Besides it takes around 0.7 s to detect each frame, which means that the fps is more than 1. Due to the limited computing power, the embedded device is still insufficient to run the face detector in real time.

On the other side, RetinaFace detector has a much shorter running time on a desktop computer with more computing power. We applied the whole network on a machine with a NVIDIA GeForce RTX 2080 Ti GPU that has a total memory of 10.76 GB. This graphical processing unit has 7.5 times larger computing capacity than the one in the embedded device mentioned above. The RetinaFace detector takes an average of 0.025 s to detect one frame and the running speed reaches around 40 fps, which is far better than the real time detection on an embedded platform. However, the detection time will grow with the increase of the number of subjects in the image. For example, when there is only one subject in the image, the detector takes less than 0.02 s to detect the faces in one frame.

### 5.2. Vital Sign Estimation

To simulate the most potential use case of this study where people’s vital signs are remotely measured at the entrances (i.e., stores, buildings), the collected signals from our experiments were divided into 15 s’ segments using the sliding window method with 50% overlap between two consecutive segments for analysis.

#### 5.2.1. Respiration Rate Estimation

The mean absolute error (MAE) between the estimated RR and reference RR of each respiration cycle was employed to evaluate the performance of the proposed system. The respiratory signal examples extracted from $ROI_{RR}$ and a reference breath belt of a subject with different face masks and without face mask can be found in Figure 13. A demo video of respiration monitored by a thermal camera when a subject was wearing different face masks can be found in Appendix A. The MAE between the estimated RR and reference RR of this subject with different face masks and without face masks at different detection ranges is shown in Table 1.

For one-subject RR estimation with and without face masks at different detection ranges, the mean absolute RR estimation error for each respiration cycle were calculated for each subject and can be found in Table 2. The correlation plots and the Bland–Altman plots of the estimated RR and the reference RR for experiments when the subjects with and without face masks are shown in Figure 14. The MAE for all one-subject RR experiments is 1.52 breaths/min. For one-subject experiments, when subjects were not wearing face masks, the MAE is 1.83 breaths/min and the Pearson correlation coefficient between the estimated RR and reference RR is 0.83 as shown in Figure 14a. The mean bias is −0.012 breaths/min and the limits of agreement are −4.63 breaths/min and 4.40 breaths/min as demonstrated in Figure 14b. For one-subject experiments, when subjects were wearing face masks, the MAE is 1.21 breaths/min and the Pearson correlation coefficient between the estimated RR and reference RR is 0.91 as shown in Figure 14c. The mean bias is −0.24 breaths/min and the limits of agreement are −3.45 breaths/min and 2.96 breaths/min as demonstrated in Figure 14d.

It can be observed that the overall performance of RR estimation for experiments with face masks is better than experiments without face masks. This is because the area of temperature changes around the nostril is larger and the temperature changes caused by respiration are more obvious when subjects are wearing face masks.

#### 5.2.2. Heart Rate Estimation

The absolute difference (eHR) between the estimated HR and reference HR of each cardiac cycle was employed to evaluate the performance of the proposed system. The mean ($M_{eHR}$) and standard deviation ($SD_{eHR}$) of the absolute HR estimation error for each cardiac cycle are calculated for each subject different distances, as shown in Table 3. The correlation plots and the Bland-Altman plots of the estimated HR and the reference HR for subjects with different skin tones are shown in Figure 15. The MAE for all one-subject HR experiments is 2.79 beats/min. For subjects who have a pale skin tone, the MAE is 2.87 beats/min and the Pearson correlation coefficient between the estimated HR and reference HR is 0.96 as shown in Figure 15a. The mean bias is 0.17 beats/min and the limits of agreement are −7.21 beats/min and 7.56 beats/min as demonstrated in Figure 15b. For subjects who have medium or dark skin tones, the MAE is 2.66 beats/min and the Pearson correlation coefficient between the estimated HR and reference HR is 0.97 as shown in Figure 15c. The mean bias is 0.25 beats/min and the limits of agreement are −6.53 beats/min and 7.03 beats/min as demonstrated in Figure 15d.

#### 5.2.3. Two-Subjects RR and HR Estimation

The feasibility of estimating multiple subjects’ RR and HR at the same time using our proposed methods was verified. Two subjects with and without face masks were required to stand in front of the cameras for two minutes at a distance of 70 cm and 100 cm, respectively. The breathing belts were used to collect reference respiratory waveform from their chests and PPG sensors were used to measure reference HR from their fingertips. The mean and standard deviation of the absolute errors between the estimations and references for each experiment are demonstrated in Table 4.

## 6. Conclusions

This paper presented a technique for contactless measurements of three vital signs of a subject including forehead temperature, respiratory rate and heart rate at the same time. We designed the system based on one thermal camera and one RGB camera. Ten healthy subjects with different skin colors participated in several different experiments to verify the feasibility of the proposed system for real-world applications. By comparing with the reference data collected at the same time from other devices, such as a thermometer, a respiration chest-belt and a PPG sensor, it was shown that our methods have acceptable accuracy in estimating BT, RR and HR.

Our methods can detect multiple subjects in a very short time, and can handle the problem of faces being occluded by masks very well. Our methods perform well in estimating vital signs at near distances (60 cm–120 cm) for subjects with and without face masks, which are great improvements compared to previous research because we enhance the detection range and allow for the wearing of masks and slight movement of the subjects during the measurements. As a result, our system is capable of contactless monitoring of vital signs and can be potentially applied for the entrance screening of people’s health condition. For application scenarios, such as detecting the vital signs in an emergency situation, HR measurement with an error less than 5 beats/min is likely to be acceptable [7].

For future work, we will implement our methods on the embedded devices or mobile devices to make the whole system portable and capable of running in real time, and a larger study population will be included to validate the stability and repeatability of the proposed approach. In addition, we will use our methods on other thermal cameras and RGB cameras to test the robustness and portability of the methods.

## Figures and Tables

**Figure 1 sensors-22-00627-f001:**
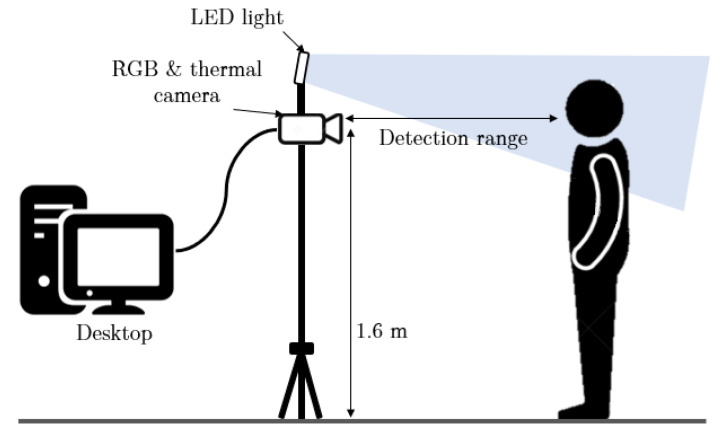
The illustration of vital signs measurement experiment setup.

**Figure 2 sensors-22-00627-f002:**
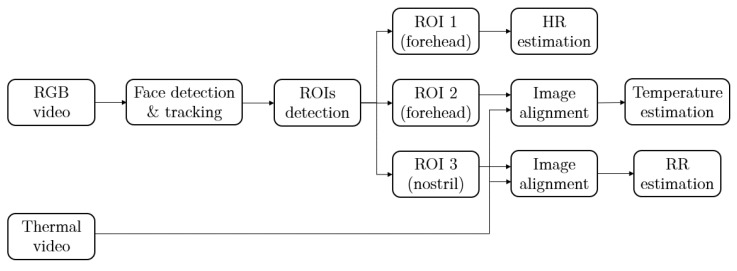
The workflow of vital signs estimation using RGB and thermal cameras.

**Figure 3 sensors-22-00627-f003:**
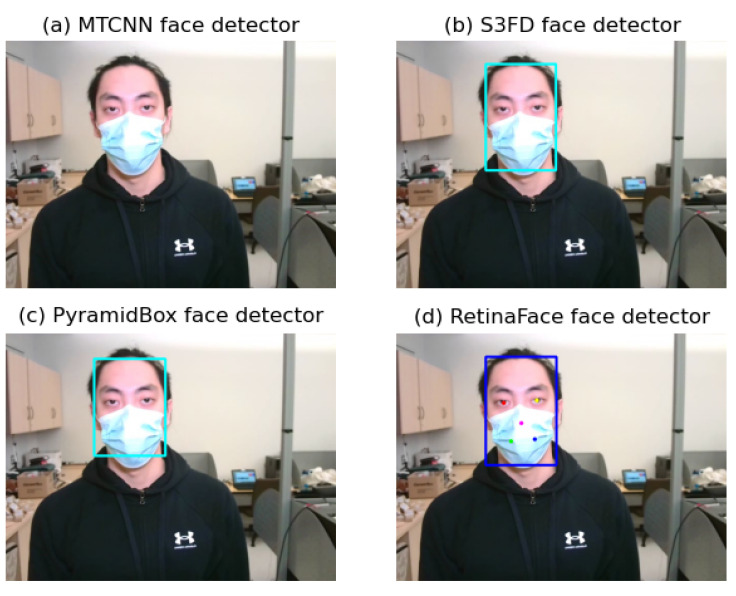
An example of face detection using different detectors (**a**). No face detected by MTCNN face detector, (**b**) face (blue box) detected by S3FD face detector, (**c**) face (blue box) detected by PyramidBox face detector, (**d**) face (blue box) and facial landmarks (colored dots) detected by RetinaFace detector.

**Figure 4 sensors-22-00627-f004:**
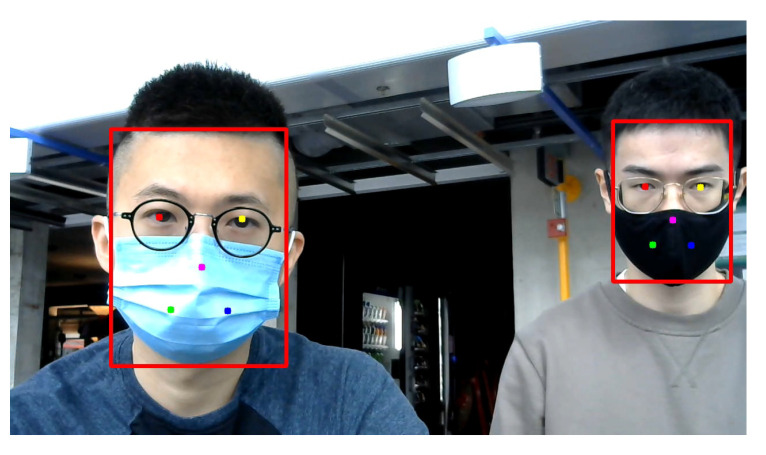
An example of multiple subjects’ faces (red bounding boxes) and facial landmarks (colored dots) detected by RetinaFace detector when they were wearing face masks.

**Figure 5 sensors-22-00627-f005:**
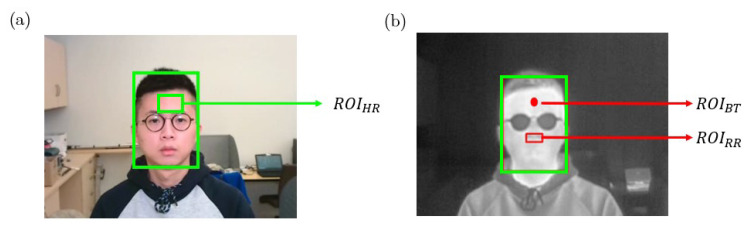
ROIs employed for vital signs estimation, (**a**). the location of $ROI_{HR}$ on the RGB image, (**b**). The locations of $ROI_{RR}$ and $ROI_{BT}$ on the simultaneous thermal image.

**Figure 6 sensors-22-00627-f006:**
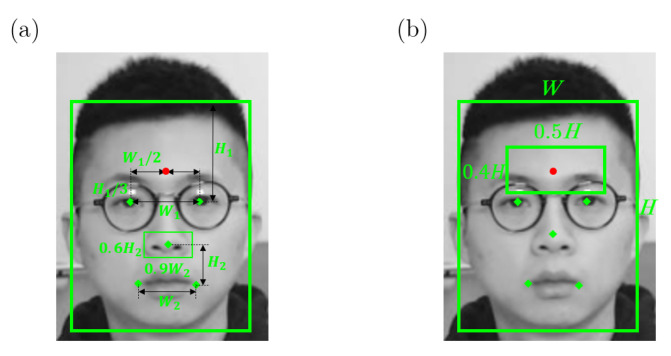
A demonstration of ROIs localization using facial landmarks, (**a**) the definitions of $ROI_{BT}$ and $ROI_{RR}$, (**b**) the definition of $ROI_{HR}$.

**Figure 7 sensors-22-00627-f007:**
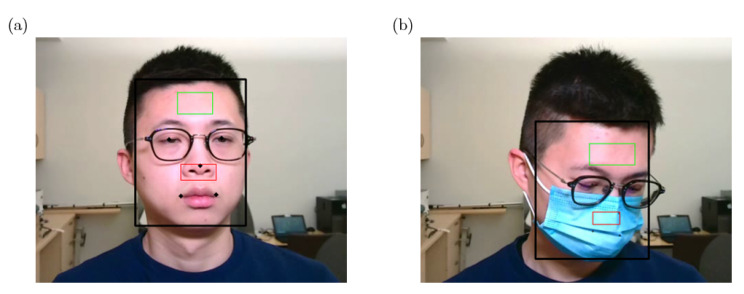
(**a**) Facial landmarks and partitions when the subject is looking at the camera. (**b**) Scenario when the subject is not looking at the camera and the ratio $Face_{deflection}$ reaches 0.17.

**Figure 8 sensors-22-00627-f008:**
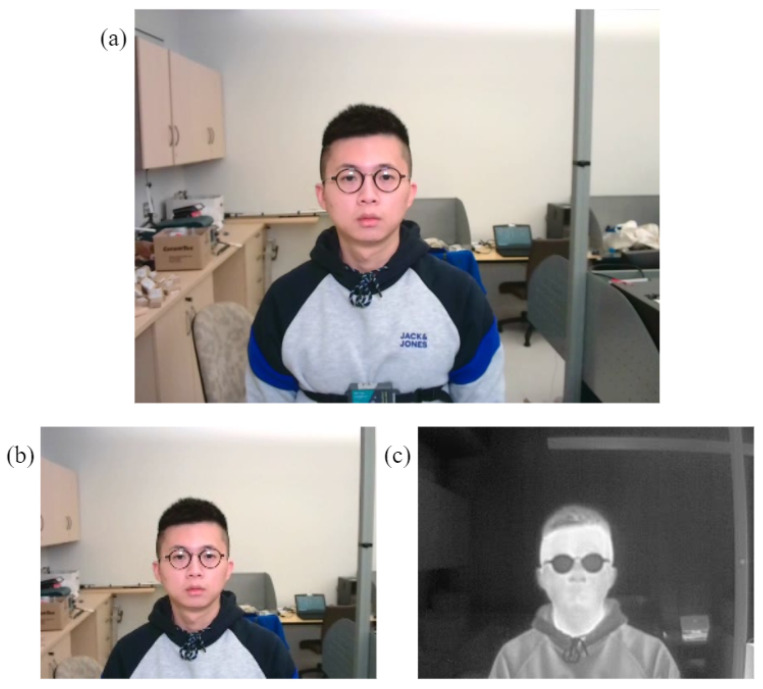
Demonstration of RGB and thermal image alignment, (**a**) original RGB image (640 × 480), (**b**) cropped RGB image (256 × 192), (**c**) aligned thermal image (256 × 192).

**Figure 9 sensors-22-00627-f009:**
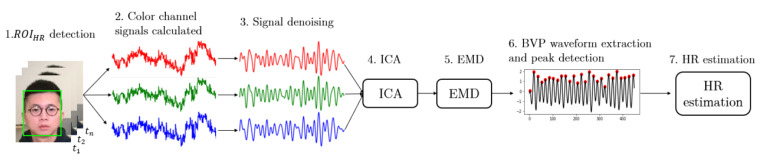
Methods developed for remote HR estimation.

**Figure 10 sensors-22-00627-f010:**
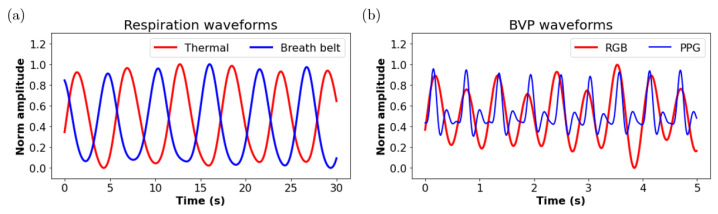
(**a**) Respiration waveform extracted from the thermal camera (red) and the reference measurement using a breath belt (blue), (**b**) BVP waveform extracted the RGB camera (red) and reference PPG waveform (blue).

**Figure 11 sensors-22-00627-f011:**
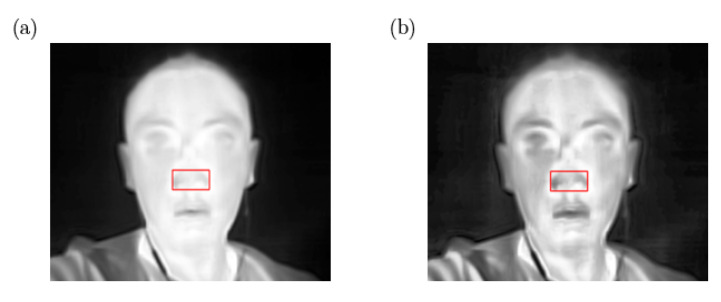
(**a**) Original thermal image with $ROI_{RR}$ detected (red bounding box), (**b**) the contrast of the thermal image enhanced by CLAHE.

**Figure 12 sensors-22-00627-f012:**
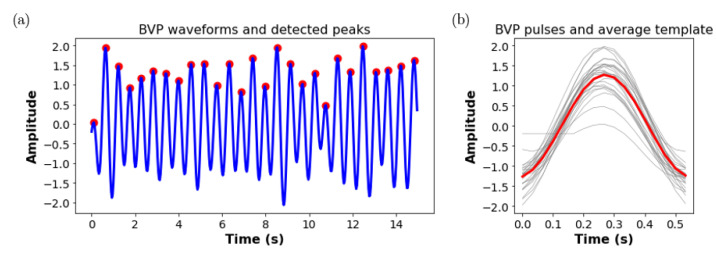
Demonstration of signal quality assessment, (**a**) A 15s rBVP waveform and detected peaks using proposed approaches, (**b**) rBVP pulses (gray curves) and the averaged rBVP template (red curves), SQI = 0.94.

**Figure 13 sensors-22-00627-f013:**
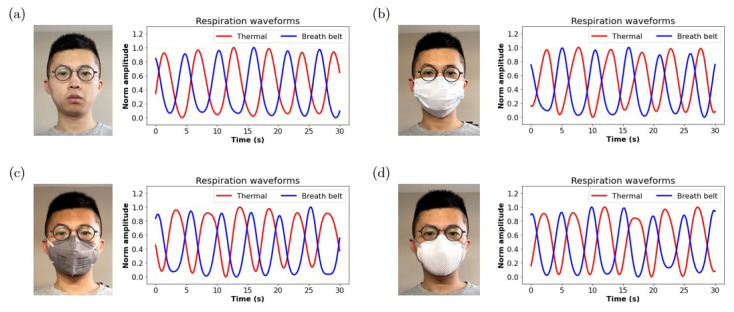
Respiratory signals extracted from thermal video (red) and from a reference breath belt (blue) of a subject at 75 cm with different face masks, (**a**) No face mask, (**b**) medical face mask, (**c**) N90 face mask, and (**d**) N95 face mask.

**Figure 14 sensors-22-00627-f014:**
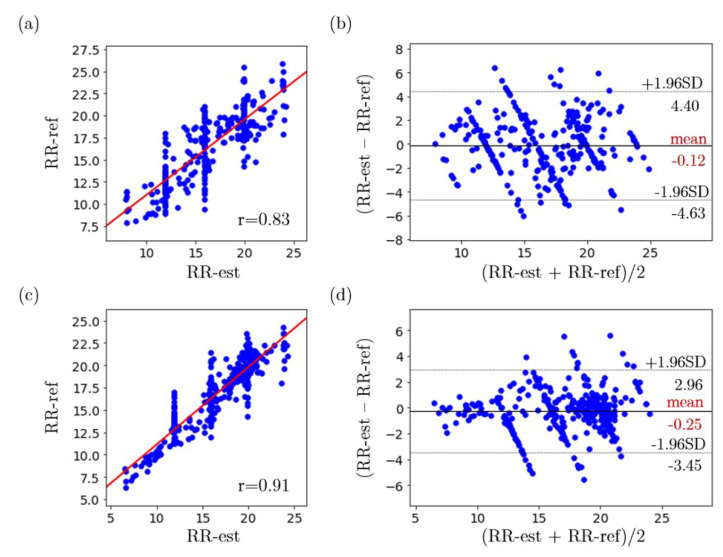
(**a**) Correlation plot of the estimated RR and reference RR for experiments when subjects were not wearing face masks, (**b**) Bland–Altman plot of the estimated RR and reference RR for experiments when subjects were not wearing face masks, (**c**) Correlation plot of the estimated RR and reference RR for experiments when subjects were wearing face masks, (**d**) Bland–Altman plot of the estimated RR and reference RR for experiments when subjects were wearing face masks.

**Figure 15 sensors-22-00627-f015:**
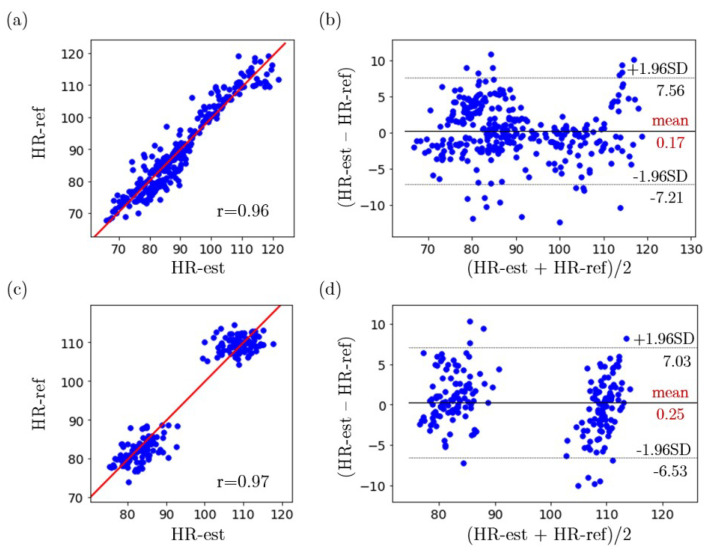
(**a**) Correlation plot of the estimated HR and reference HR for subjects who have a pale skin tone, (**b**) Bland–Altman plot of the estimated HR and reference HR for subjects who have a pale skin tone, (**c**) Correlation plot of the estimated HR and reference HR for subjects who have medium or dark skin tones, (**d**) Bland–Altman plot of the estimated HR and reference HR for subjects who have medium or dark skin tones.

**Table 1 sensors-22-00627-t001:** The mean absolute error between the estimated RR and the reference RR of a subject with different face masks at different detection ranges (unit: breaths/min).

	75 cm	120 cm	200 cm
No mask	0.41	0.28	0.80
Mask1 (medical/cloth)	0.52	0.45	0.59
Mask2 (N90)	0.06	0.01	0.17
Mask3 (N95)	0.06	0.42	0.88

**Table 2 sensors-22-00627-t002:** Mean absolute error between the estimated RR and the reference RR (unit: breaths/min). **Boldface** character denotes the best result.

	60 cm		80 cm		100 cm		120 cm	
	No Mask	Mask	No Mask	Mask	No Mask	Mask	No Mask	Mask
Subject1	1.83	0.50	2.04	1.60	2.73	0.74	2.08	0.74
Subject2	2.68	0.40	2.24	0.36	2.49	0.23	1.25	0.52
Subject3	1.84	0.46	1.72	0.56	1.69	0.57	2.01	**0.18**
Subject4	1.70	2.30	2.06	1.65	2.31	1.69	2.09	1.54
Subject5	1.04	1.65	1.99	1.75	2.10	0.88	1.26	1.15
Subject6	1.94	2.22	2.58	1.40	2.45	1.04	3.10	2.16
Subject7	1.82	0.59	2.18	0.49	0.65	0.60	1.14	0.78
Subject8	1.65	1.24	1.35	0.64	2.17	1.01	1.38	1.14
Subject9	1.18	1.67	0.99	0.96	1.12	1.07	1.24	1.42
Subject10	1.71	2.47	2.30	1.80	2.85	2.46	1.56	1.70

**Table 3 sensors-22-00627-t003:** Mean and standard deviation of the absolute error between the estimated HR and the reference HR (unit: beats/min). **Boldface** character denotes the best result.

		60 cm		80 cm		100 cm		120 cm	
	Skin Tone	$M_{\mathrm{eHR}}$	${\mathrm{SD}}_{\mathrm{eHR}}$	$M_{\mathrm{eHR}}$	${\mathrm{SD}}_{\mathrm{eHR}}$	$M_{\mathrm{eHR}}$	${\mathrm{SD}}_{\mathrm{eHR}}$	$M_{\mathrm{eHR}}$	${\mathrm{SD}}_{\mathrm{eHR}}$
Subject1	pale	2.92	1.21	3.02	1.11	4.31	1.57	2.93	1.31
Subject2	pale	3.58	2.59	4.08	3.26	3.82	3.05	4.83	4.00
Subject3	pale	2.61	2.42	4.28	2.86	3.89	3.01	4.50	2.88
Subject4	pale	3.35	2.31	2.17	1.82	2.88	1.82	2.50	1.23
Subject5	pale	1.73	3.21	2.62	1.44	1.81	1.76	2.03	1.65
Subject6	pale	1.50	1.78	**1.05**	**1.43**	1.86	2.29	1.76	3.02
Subject7	medium	1.68	1.55	2.61	2.90	3.05	2.55	3.09	2.69
Subject8	medium	2.53	1.53	3.07	3.05	2.35	2.59	3.04	2.89
Subject9	dark	2.72	1.75	3.83	1.88	3.31	2.44	2.21	1.56
Subject10	dark	3.38	2.15	2.20	1.86	2.20	1.50	2.09	1.88

**Table 4 sensors-22-00627-t004:** Mean and standard deviation of the absolute error of two-subjects RR and HR estimations (RR unit: breaths/min, HR unit: beats/min).

	Subject1		Subject2	
	RR	HR	RR	HR
Experiment 1	0.89 ± 0.47	3.60 ± 2.10	1.31 ± 0.86	1.72 ± 1.40
Experiment 2	1.25 ± 0.63	2.17 ± 2.20	0.80 ± 0.57	2.41 ± 1.13
Experiment 3	0.49 ± 0.33	1.79 ± 1.06	1.07 ± 0.58	1.35 ± 1.12
Experiment 4 (mask)	1.57 ± 0.96	2.03 ± 1.18	0.74 ± 0.52	2.39 ± 1.67
Experiment 5 (mask)	1.60 ± 0.58	2.15 ± 2.07	0.49 ± 0.32	2.31 ± 1.09

## Data Availability

The data presented in this study are available on request from the corresponding author. The data are not publicly available due to data sharing is not included in initial project proposal.

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
