# Peer review of "Contactless Measurement of Vital Signs Using Thermal and RGB Cameras: A Study of COVID 19-Related Health Monitoring"

_sensors, 2022, doi:10.3390/s22020627_

Round 1

Reviewer 1 Report

The paper is really interesting and well written. Moreover the procedure is explained consistently and almost flawless. Authors used a number of well established techniques to address the problem of measuring a number of features in real-time. My only concerns are on the scientific novelty of the proposedmethod. So I am asking authors to provide more explaination in the introduction regarding the main innovative aspects proposed by their work.

Reviewer 2 Report

The authors presented contactless methods for simultaneously recording three vital signs such as forehead temperature, respiratory rate, and heart rate of the subject. The system is based on a thermal camera and an RGB camera. The manuscript is well written and should be great attention to the readers. However, I have many concerns about the style of this paper and provided information about the topic. The proposal of this work presents several deficiencies that need to be addressed.

In the 3rd point of the contribution, the authors said address some issue related to ‘small distance required between the cameras ….’ It conflicts with the later statement. The authors should make it clear for better understanding. 

The authors conducted experiments on a limited dataset, so authors are advised to conduct more experiments on large data. The sample size should be more than 30 subjects in order to satisfy the condition of statistical significance.

A detailed performance evaluation must be presented. The authors should compare their proposed method results with at least three most recently published works to show the novelty of this paper.

Using ICA and EMB in your work is not a contribution unless the authors invent these two algorithms. Furthermore, the authors should give one paragraph to describe why ICA and EMD are important/needed in this work. What is the goal to perform ICA and EMD? There are several different methods to remove noise or perform dimensionality reduction, such as PCA, LDA, and low-pass filters.

Almost in all tables, the mean absolute error rate is lower with the mask and higher without a mask. Such results contradict to our common sense. The score of mean absolute error should be lowest in the case of without using a mask, and with a mask, it should be higher.

Round 2

Reviewer 2 Report

The authors cannot provide satisfactory answers to the raised issues. There is no novelty in the paper.  
